# Reduction of Ethyl Carbamate in an Alcoholic Beverage by CRISPR/Cas9-Based Genome Editing of the Wild Yeast

**DOI:** 10.3390/foods12010102

**Published:** 2022-12-25

**Authors:** Jin-Young Jung, Min-Ji Kang, Hye-Seon Hwang, Kwang-Rim Baek, Seung-Oh Seo

**Affiliations:** Department of Food Science and Nutrition, The Catholic University of Korea, Bucheon 14662, Republic of Korea

**Keywords:** ethyl carbamate, alcoholic beverages, *Saccharomyces cerevisiae*, CRISPR/Cas9, genome editing

## Abstract

Ethyl carbamate (EC) is a naturally occurring substance in alcoholic beverages from the reaction of ethanol with urea during fermentation and storage. EC can cause dizziness and vomiting when consumed in small quantities and develop kidney cancer when consumed in excess. Thus, the reduction of EC formation in alcoholic beverages is important for food safety and human health. One of the strategies for reducing EC contents in alcoholic beverages is developing a new yeast starter strain to enable less formation of EC during fermentation. In this study, we isolated a polyploid wild-type yeast *Saccharomyces cerevisiae* strain from the *Nuruk* (Korean traditional grain-based inoculum of wild yeast and mold) and developed a starter culture by genome engineering to reduce EC contents in alcoholic beverages. We deleted multiple copies of the target genes involved in the EC formation in the *S. cerevisiae* by a CRISPR/Cas9-based genome editing tool. First, the *CAR1* gene encoding for the arginase enzyme responsible for the formation of urea was completely deleted in the genome of *S. cerevisiae*. Additionally, the *GZF3* gene encoding the transcription factor controlling expression levels of several genes (*DUR1*, *2*, and *DUR3*) related to urea absorption and degradation was deleted in *S. cerevisiae* to further reduce the EC formation. The effects of gene deletion were validated by RT-qPCR to confirm changes in transcriptional levels of the EC-related genes. The resulting strain of *S. cerevisiae* carrying a double deletion of *CAR1* and *GZF3* genes successfully reduced the EC contents in the fermentation medium without significant changes in alcohol contents and fermentation profiles when compared to the wild-type strain. Finally, we brewed the Korean traditional rice wine *Makgeolli* using the double deletion strain of *S. cerevisiae* dCAR1&GZF3, resulting in a significant reduction of the EC content in *Makgeolli* up to 41.6% when compared to the wild-type strain. This study successfully demonstrated the development of a starter culture to reduce the EC formation in an alcoholic beverage by CRISPR/Cas9 genome editing of the wild yeast.

## 1. Introduction

Ethyl carbamate (EC) is a naturally occurring substance during alcoholic fermentation by the reaction of urea with ethanol [1]. EC is known as a carcinogen in experimental animals and possibly carcinogenic to humans (IARC group 2A) [2]. Consumption of EC can cause unconsciousness, vomiting, and damage to the liver and kidneys [1]. As estimated by the JECFA (Joint FAO/WHO Expert Committee on Food Additives), the benchmark dose lower limit of ethyl carbamate is 0.3 mg/kg body weight per day [1]. EC is detected in many alcoholic beverages including beer, wine, brandy, and others at concentrations ranging from 0.01 to 12 mg/L depending on the origin of spirit [1,3]. Considering the high EC contents in alcoholic beverages, several countries have limitations on their levels [1]. For example, maximum levels for ethyl carbamate in wine were limited to 30 μg/L in table wine, 150 μg/L in distilled spirit, and 400 μg/L in fruit brandy in Canada [1]. The reduction of EC concentration in alcoholic beverages is important for ensuring food safety and quality control.

Since EC is constantly produced during fermentation and storage of alcoholic beverages, many efforts have been made to reduce EC contents in fermented foods. First, fermentation temperature can be adjusted to reduce EC formation during yeast fermentation. Second, urease enzyme can be added to the fermentation medium to directly eliminate urea, the major precursor for EC formation. However, urea can be continuously produced from the arginine metabolism of the brewer’s yeast *Saccharomyces cerevisiae* during fermentation. To reduce urea formation, yeast starter strain can be metabolically engineered to modify the arginine metabolism (Figure 1). Arginine is cleaved into ornithine and urea by arginase encoded by the *CAR1* gene in *S. cerevisiae*. In the previous study, the *CAR1* gene was deleted in the *S. cerevisiae* lab strain, resulting in EC reduction in the growth medium [4]. Additionally, *DUR1, 2*, and *DUR3* genes involved in urea degradation were overexpressed in the Sake yeast to reduce the urea contents [5]. One study attempted to reduce urea and EC in rice wine through modification of expression levels of four GATA transcriptional factors (*GLN3*, *GAT1*, *DAL80*, and *GZF3*) related to nitrogen catabolite repression in *S. cerevisiae* [6]. Disruption of the GZF3 transcription factor showed the most effective results for urea and EC reduction in model rice wine [6]. If the industrial strain of *S. cerevisiae* strain currently used for brewing can be safely engineered for ethyl carbamate reduction, the strain can be directly used for industrial scale production of alcoholic beverages with reduced ethyl carbamate contents.

In this study, we aimed to delete both *CAR1* and *GZF3* genes in the industrial *Saccharomyces cerevisiae* strain isolated from Korean traditional fermentation ingredients *Nuruk* containing various wild yeast and molds, and brew alcoholic beverages with low EC contents by the engineered *S. cerevisiae* (Figure 1). To relieve the regulatory and safety issues of GMO products, we engineered the *S. cerevisiae* using the CRISPR/Cas9 genome editing tool which is useful to delete the multiple genes in the genome of target microorganism safely without introducing antibiotics resistant markers or plasmids. Since EC is detected in *Makgeolli*, a traditional Korean rice wine fermented by *Saccharomyces cerevisiae* [7], we demonstrate the reduction of EC contents in *Makgeolli* by brewing with the engineered *S. cerevisiae*. The EC and ethanol contents in *Makgeolli* fermented by the engineered *S. cerevisiae* and its parental strain were evaluated. The impact of gene deletions on the expression levels of the EC-related genes in *S. cerevisiae* was also investigated.

## 2. Materials and Methods

### 2.1. Materials

The chemical used in the experiment was purchased from Sigma-Aldrich (St. Louis, MO, USA) and media was purchased from Thermo Fisher Scientific (Waltham, MA, USA). The yeast strain was collected from the variable regions of South Korea. *Aspergillus oryzae* Enzyme Powder^™^ was purchased from JEIL BIOTECH Co. (Gyeongju, South Korea) and *Japonica* rice used from Icheon rice (Icheon, South Korea). All strains and plasmids used in this study are summarized in Table 1. All primers used in this study are summarized in Table 2.

### 2.2. Strains and Plasmids

The *Escherichia coli* TOP10 strain was used for the construction and propagation of plasmids. The *S. cerevisiae* GRL6 strain isolated from *Nuruk* was used for the construction of EC reduction yeast. To isolate yeast strain in *Nuruk*, 1 g of *Nuruk* was mixed with 100 mL of DDW and serially diluted. The serially diluted sample was spread onto YPD2 agar (5 g/L yeast extract, 10 g/L peptone, 2 g/L glucose, 1.8 g/L agar) plate and the colonies formed after 2 days of incubation at 30 °C [9]. A single colonization was taken and observed under a microscope to find a group of yeast candidates. The colony PCR was performed using the colony of the candidate group for identification. The ITS1 and ITS4 primers were used to amplify the internal transcribed spacer region of the 18S rRNA gene [10]. The sequence of refined PCR products was sent to BIONICS (Seoul, South Korea) for Sanger sequencing. The BLAST (http://www.ncbi.nlm.nih.gov/BLAST/ (accessed on 18 March 2020) of the National Center for Biotechnology Information (NCBI, MD, USA) was used to obtain homology information [11].

The pCas9-Hyg plasmid containing hygromycin B antibiotics marker was constructed to express the Cas9 protein in yeast. The PCR fragment was amplified from the pCas9-NAT plasmid except for Nourseothricin antibiotics marker region by the Cas9-NAT-backbone-F and Cas9-NAT-backbone-R primers. The Hygromycin B-resistant marker was amplified from the pRS42H-HXK2 plasmid by the Hyg-insert-F and Hyg-insert-R primers. These two PCR fragments were assembled by the Gibson Assembly^®^ Cloning Kit from NEB (Ipswich, MA, USA) to construct the pCas-Hyg plasmid.

The pRS42K-gRNA-*CAR1* and pRS42K-gRNA-*GZF3* were constructed for gRNA expression to target *CAR1* and *GZF3* genes to be deleted. The construction of pRS42K-gRNA-*CAR1* and pRS42K-gRNA-*GZF3* used the Q5^®^ Site-Directed Mutagenesis Kit from NEB (Ipswich, MA, USA). Using Site-Directed Mutagenesis, the gRNA targeting 20 bp of the *HXK2* gene in the pRS42H-HXK2 was replaced to target the *CAR1* and *GZF3* genes, respectively. The PCR fragment for the construction of the pRS42K-gRNA-*CAR1* plasmid was amplified using the SDM-gRNA-*CAR1*-F, and SDM-gRNA-*CAR1*-R primers. The PCR fragment for pRS42K-gRNA-*GZF3* construction was amplified using the SDM-gRNA-GZF3-F, and SDM-gRNA-GZF3-R primers. The detailed experimental methods followed the Q5^®^ Site-Directed Mutagenesis Kit Protocol.

### 2.3. Media and Culture Conditions

*E. coli* was grown in Luria–Bertani (LB) medium (5 g/L yeast extract, 10 g/L tryptone, 10 g/L NaCl) at 37 °C, and ampicillin (50 μg/mL) was added when required. *S. cerevisiae* was grown in YPD2 medium (5 g/L yeast extract, 10 g/L peptone, 2 g/L glucose) at 30 °C and Nourseothricin sulfate (ClonNAT, 100 µg/mL), Geneticin (G418, 300 µg/mL) were added when required. For comparison of EC formation, batch fermentation was performed in the YPD8 medium (5 g/L yeast extract, 10 g/L peptone, 80 g/L glucose) at pH 6.8 and 30 °C. Arginine 50 mM and urea 50 mg/L which are similar levels to the raw ingredients of alcoholic beverages were added to the medium to provide the EC formation condition [12,13].

### 2.4. Chromosomal Gene Deletion by CRISPR/Cas9 System

After the DNA double-strand break using the CRISPR/Cas9 system, the *S. cerevisiae* GRL6 strain was repaired by the 90 bp donor DNA through homologous recombination. The donor DNA of *CAR1* gene deletion was constructed by the Donor-*CAR1*-F and Donor-*CAR1*-R primers. The donor DNA of *GZF3* gene deletion was constructed by the Donor-*GZF3*-F and Donor-*GZF3*-R primers. Each 60 bp forward and reverse primer contains a complementary 30 bp homologous region.

The pCas9-Hyg, pRS42K-gRNA-*CAR1,* and the *CAR1* donor DNA were transformed into *S. cerevisiae* GRL6 for the *CAR1* gene deletion. The pCas9-Hyg, pRS42K-gRNA-*GZF3,* and the *GZF3* donor DNA were transformed into *S. cerevisiae* GRL6 and *S. cerevisiae* Δ*CAR1* for *GZF3* gene deletion, respectively. The transformation was performed by the high-efficiency yeast transformation method using the LiAc/SS carrier DNA/PEG [14]. After the transformation, the gene deletion was confirmed by the colony PCR. The *CAR1* gene deletion was checked by the Check-CAR1-F and Check-CAR1-R primers (WT: 3112 bp, *CAR1* deletion: 2100 bp) and the *GZF3* gene deletion was checked by the Check-GZF3-F and Check-GZF3-R primers (WT: 2756 bp, *GZF3* deletion: 900 bp).

### 2.5. Flow Cytometry Analysis of Yeast Ploidy

The method referred to the Propidium Iodide Nucleic Acid Stain Manuals by Invitrogen (Waltham, MA, USA) and previous ploidy analysis studies [15,16]. The yeast cells were inoculated into 5 mL of YPD2 medium (20 g/L glucose) at 30 °C. When the optical density of the culture reached to 1.0 (OD_600_), the cells were collected and resuspended in 500 μL of water and centrifuged for 3 min (500× *g*). After removing the supernatant, 1 mL of phosphate-buffered solution (PBS) at room temperature and 4 mL absolute ethanol at −20 °C were added to the cell pellet and slowly resuspended. After vortexing of the cell suspension, the cells stayed in ethanol at −20 °C for 20 min. Then, the solution was centrifuged for 1 min (3000× *g*), and the cells were harvested and rehydrated by adding 5 mL of PBS for 15 min. Rehydrated cells were centrifuged at 3 min (3000× *g*) and the supernatant was removed. One mL of 3 μM Propidium Iodide (PI) solution in buffer (100 mM Tris, pH 7.4, 1 mM CaCl_2_, 150 mM NaCl, 0.5 mM MgCl_2_, 0.1% Nonidet P-40) was added and slowly resuspended. The samples stayed for 1 h at room temperature in the dark. The stained cells were stored at 4 °C until analysis. The samples were briefly sonicated and analyzed with Cytek^®^ Biosciences Flow Cytometry Analyzer (NL-1000) using a Texas Red filter. Histograms were acquired in linear mode. The data were analyzed with SpectroFlo^®^ software (Cytek Biosciences, Fremont, CA, USA). 

### 2.6. RNA Preparation and Real-Time Quantitative PCR Assay

We collected yeast cells grown for 6 h in batch fermentation and used them for RNA preparation and Real-Time Quantitative PCR (RT-qPCR) Assay. The preparation of yeast RNA was performed by *AccuPrep*^®^ Universal RNA Extraction Kit from BIONEER (Daejeon, Korea). The detailed experimental methods followed the *AccuPrep*^®^ Universal RNA Extraction Kit Protocol. The cDNA synthesis of yeast RNA was carried out using the PrimeScript^™^ 1st strand cDNA Synthesis Kit from Takara Bio (Shiga, Japan) and its protocol.

The RT-qPCR was performed to determine the RNA expression using a Real-Time PCR instrument (Applied Biosystems, CA, USA). The parameters for PCR were as follows: pre-incubation at 95 °C for 30 s, then 40–50 cycles of amplification at 95 °C for 5 s, 55 °C for 20 s, and cooling at 50 °C for 30 s. The relative expression of the *CAR1, DUR1, 2,* and *DUR3* genes was determined by the ΔΔCt method based on the *ACT1* gene which is known to be continuously expressed [17,18]. Primers used for RT-qPCR analysis are summarized in Table 2.

### 2.7. Brewing Korean Rice Wine Makgeolli

The Korean rice wine *Makgeolli* was brewed to measure the EC production during the fermentation by the engineered yeasts and its parent strain. The 100 g of *japonica* rice was steamed for 30 min and cooled. The yeast was cultivated in YPD2 medium overnight at 30 °C. For brewing of *Makgeolli*, 300 mL of water, steamed rice, 10 g of *Aspergillus oryzae* enzyme powder, and yeast cells OD_600_ = 50 (around 15 g DCW, and 0.7 × 10^9^ cells) were mixed. Fermentation was performed for 10 days at 20 °C by stationary culture.

### 2.8. Fermentation Capability and EC Analysis

The optical density of yeast culture was measured at 600 nm absorbance using a UV-1800 spectrophotometer (SHIMADZU, Kyoto, Japan). For analysis of fermented products, culture supernatants were diluted appropriately after centrifugation of the batch fermentation medium and *Makgeolli*. Concentrations of glucose, glycerol, acetate, and ethanol were measured using the 1260 Infinity II HPLC system of Agilent Technologies (Santa Clara, CA, USA) equipped with an Aminex^®^ HPX-87H column (300 × 7.8 mm) (Bio-Rad, Hercules, CA, USA). The mobile phase was 5 mM H_2_SO_4_ solution and the flow rate was 0.6 mL/min. The column was heated at 60 °C to analyze 10 μL of a diluted supernatant sample using the previous method [4].

For EC quantification, standards and samples were prepared according to a previous study [19]. The analysis of EC content was performed using the 6890 N gas chromatograph (Agilent Technologies, CA, USA) equipped with the 5973 Mass Selective Detector (GC-MS) (Agilent Technologies, CA, USA). The samples were separated on a DB-5 MS column (30 m × 0.250 mm ID × 0.25 μm) (Agilent Technologies, CA, USA). The flow rate of the helium carrier gas was 0.5 mL/min, and 1 μL of the extracts was injected with splitless mode. The injector temperature was 200 °C and the oven was programmed as following conditions (50 °C held for 2 min, 2 °C/min increased to 100 °C, and 30 °C/min increased to 200 °C, held for 10 min). The mass spectral identification of compounds was applied at an ion source temperature of 230 °C and mass ionization voltage of 70 eV. The selected ion monitoring (SIM: m/z 62, 74, 89) was used for the analysis of EC. For quantitative analysis of EC, m/z 62 which is the major fragment ion was used and has few interruptions between fragments [20]. 

### 2.9. Statistical Analysis

At least three repeat experiments were performed for each fermentation experiment. The results were expressed as average with standard errors. All statistical analyses were performed using SAS 9.4 from SAS Institute Inc. (Cary, NC, USA). Data were analyzed using One-way ANOVA. *p*-values of less than 0.05 were considered statistically significant.

## 3. Results and Discussion

### 3.1. Isolation of Yeast Stater Strain and Ploidy Analysis

To develop a starter culture to reduce EC contents in alcoholic beverages, we first isolated the wild-type *Saccharomyces cerevisiae* strain from the *Nuruk,* Korean traditional grain-based inoculum of wild yeast and mold. The *Nuruk* sample was diluted with water and spread onto the YPD2 agar plate. After 2 days of incubation, several hundred colonies were grown on the plate. Among them, a single colony was selected based on ethanol production phenotype. The isolated colony named GRL6 was observed under a microscope and showed a yeast-like shape. For identification of the isolated strain, the colony PCR of the GRL6 strain was performed to check the internal transcribed spacer region of the 18S rRNA gene. Sanger sequencing and BLAST analysis suggested that the isolated strain GRL6 was identified as *S. cerevisiae*.

To confirm the ploidy of the isolated wild-type *S. cerevisiae* GRL6 from *Nuruk*, we performed a ploidy analysis of the cells stained with PI by flow cytometry. A well-known laboratory haploid strain, *S. cerevisiae* BY4742 which is a well-known laboratory haploid strain was used to compare ploidy [21,22]. Based on the analysis, *S. cerevisiae* GRL6 was estimated as a diploid strain when compared to the haploid *S. cerevisiae* BY4742 (Figure 2). This result suggested that the isolated strain of *S. cerevisiae* GRL6 has multiple copies of genes in the chromosome of the cells. Since the CRISPR/Cas9 system can be utilized for the simultaneous deletion of multi-copy genes in the chromosomes, we decided to delete several genes related to the EC formation by the Cas9 genome editing tool.

### 3.2. Deletion of CAR1 and GZF3 in Industrial Yeast Strain using CRISPR/Cas9 System

To reduce EC production in alcoholic beverages by developing a new starter culture, we attempted to delete the *CAR1* and *GZF3* genes in the newly isolated wild-type yeast strain, *S. cerevisiae* GRL6. Since the *CAR1* gene encodes the arginase enzyme responsible for the formation of urea in *S. cerevisiae,* the *CAR1* gene can be deleted for EC reduction. Additionally, the *GZF3* gene encoding the transcription factor controlling expression levels of the *DUR1*, *2* (urea amidolyase) and *DUR3* (urea active transporter) genes related to urea absorption and degradation can be deleted in *S. cerevisiae* for EC reduction phenotype [23,24] (Figure 1). We used the CRISPR/Cas9 genome editing tool to carry out the double deletion of *CAR1* and *GZF3* genes in the diploid *S. cerevisiae* GRL6 (Appendix A). 

The Cas9-Hyg plasmid was constructed and used for the Cas9 expression for target-specific double-strand DNA breakage in the genome of *S. cerevisiae*. The pRS42K-gRNA-*CAR1* and pRS42K-gRNA-*GZF3* plasmids were constructed for the sgRNA expression to guide the *CAR1* and GZF3 gene deletion by Cas9, respectively. Through the transformation of Cas9 and sgRNA expression plasmids and donor DNA for gene deletion, multi-copies of *CAR1* and *GZF3* genes were deleted in the diploid *S. cerevisiae* GRL6. As a result, we constructed the engineered *S. cerevisiae dCAR1* strain that deleted only *CAR1* genes, the *dGZF3* strain that deleted only *GZF3* genes, and the *dCAR1*&*GZF3* strain that deleted both *CAR1* and *GZF3* genes. Deletions of the CAR1 and GZF3 genes in the engineered *S. cerevisiae* strains were confirmed by colony PCR (Appendix A). The result suggests that complete deletion of the *CAR1* and *GZF3* genes in multiple chromosomes was successfully carried out in the wild-type yeast *S. cerevisiae* GRL6 strain isolated from *Nuruk*.

### 3.3. Real-Time Quantitative PCR Analysis of the Engineered S. cerevisiae

The engineered strain of *S. cerevisiae* carrying the double deletion of *CAR1* and *GZF3* genes was successfully constructed by the CRISPR/Cas9 tool in this study. To confirm the effects of gene deletion, we performed the RT-qPCR analysis and checked changes in transcriptional levels of the EC-related genes. The previous study carrying out the *GZF3* gene deletion in *S. cerevisiae* confirmed the increased RNA expression levels of *DUR1*, *2*, and *DUR3* genes [6]. Another study carrying out the *CAR1* gene deletion in *S. cerevisiae* confirmed no expression of the CAR1 through RT-qPCR [18]. Since we constructed the engineered *S. cerevisiae* strain harboring the double deletion of *CAR1* and *GZF3* genes for the first time, we performed the RT-qPCR analysis to confirm the changes in RNA expression levels of the *CAR1*, *DUR1*, *2* and *DUR3* genes in the engineered strains. The relative RNA expressions of the *S. cerevisiae dCAR1*, *dGZF3*, and *dCAR1*&*GZF3* strains suggested the significant differences in the expression levels of the *CAR1*, *DUR1*, *2*, and *DUR3* genes (Figure 3).

The expression of the *CAR1* gene was not detected in the engineered *dCAR1* and *dCAR1*&*GZF3* because the *CAR1* gene was completely deleted in these strains. Meanwhile, the engineered *S. cerevisiae dGZF3* strain exhibited the expression of the *CAR1* gene with no significant difference when compared to that of the wild-type *S. cerevisiae* GRL6. For the *DUR1*, *2* gene expression, the *dCAR1* strain showed no significant difference with the wild-type GRL6. However, the *dGZF3* and *dCAR1*&*GZF3* strains harboring the complete deletion of the *GZF3* gene showed a significant difference in the *DUR1* and *DUR2* gene expression levels when compared to the wild-type GRL6. The *DUR1* and *DUR2* gene expression levels increased by 2.12 times and 1.92 times in the *dGZF3* and the *dCAR1*&*GZF3* strains, respectively. For the *DUR3* gene expression, the *dCAR1* strain carrying only CAR1 gene deletion showed no significant difference compared to the wild-type strain. The *dGZF3* and *dCAR1*&*GZF3* strains exhibited increased *DUR3* gene expression levels by 1.41 times and 1.31 times, respectively, when compared to the wild-type *S. cerevisiae* GRL6. These results indicate that the effects of gene deletion in the engineered *S. cerevisiae* strains were successfully validated as we expected based on the previous studies. The engineered *S. cerevisiae* carrying the double deletion of *CAR1* and *GZF3* genes showed no expression of the *CAR1* and upregulation of *DUR1, 2,* and *DUR 3* expression, which may help to reduce EC formation during fermentation.

### 3.4. EC Reduction in Fermentation Medium by the Engineered S. cerevisiae

To evaluate the EC reduction and fermentation capability, batch fermentations in YPD8 medium containing 80 g/L glucose with 50 mg/L urea and 50 mM arginine were conducted by the engineered *S. cerevisiae dCAR1*, *dGZF3*, and *dCAR1*&*GZF3* strains and its parent strain GRL6 (Appendix A). The engineered *S. cerevisiae* strains did not show significant differences in fermentation profiles when compared to the wild-type GRL6. Final OD_600_ value, ethanol titer, yield, and productivity after 12 h fermentation of the engineered *S. cerevisiae* strains were similar to those of the wild-type strain (Appendix A). These results indicated that the *CAR1* and *GZF3* gene deletion did not affect the fermentation capability of *S. cerevisiae* GRL6.

Meanwhile, the EC concentrations in the batch fermentation by the engineered strains were significantly different than those of the wild-type *S. cerevisiae* GRL6 (Figure 4). At 24 h of batch fermentation, the EC formation decreased by 32.1% and 16.9% in the *dCAR1* and *dGZF3* strains, respectively. The EC concentration by the *dCAR1*&*GZF3* strain decreased up to 52.1% when compared to the wild-type GRL6. At 168 h of batch fermentation, the EC concentration decreased by 26.1% and 15.3% in the *dCAR1* and *dGZF3* strains, respectively. The *dCAR1*&*GZF3* strain harboring a double deletion of the *CAR1* and *GZF3* genes showed 33.9% of EC reduction when compared to the wild-type GRL6 (Appendix A). These results suggested that deletion of both *CAR1* and *GZF3* genes in *S. cerevisiae* showed the highest reduction in EC concentration during batch fermentation in yeast growth medium containing urea and arginine.

### 3.5. EC Reduction in Makgeolli Brewed by the Engineered S. cerevisiae

To evaluate the EC reduction in alcoholic beverages brewed by the engineered strains, we brewed *Makgeolli,* the Korean traditional rice wine using the engineered *S. cerevisiae dCAR1*, *dGZF3*, and *dCAR1*&*GZF3* strains and its parent strain GRL6 as starter cultures (Appendix A). There was no significant difference in the ethanol concentration of the *Makgeolli* products brewed by the engineered *S. cerevisiae* strains when compared to the wild-type GRL6 (Appendix A). The EC concentrations in *Makgeolli* brewed by the *dCAR1* and *dGZF3* strains slightly decreased when compared to the wild-type strain at both 24 h and 168 h (Figure 5). However, the *dCAR1*&*GZF3* strain significantly decreased the EC concentration in *Makgeolli* up to 37.0% when compared to the wild-type GRL6 strain after 24 h of the brewing (Appendix A). After 168 h, the EC concentration in *Makgeolli* brewed by the *dCAR1*&*GZF3* strain decreased by 41.6%. These results suggested that the deletion of *CAR1* and *GZF3* genes in *S. cerevisiae* helped to decrease the EC concentration in alcoholic beverages. Especially, the double deletion of *CAR1* and *GZF3* genes was most effective to reduce the EC concentration among the engineered *S. cerevisiae* strains.

## 4. Conclusions 

In this study, we successfully isolated the wild-type *S. cerevisiae* strain from *Nuruk*, the Korean traditional Fermentation ingredients used for brewing the Korean traditional rice wine *Makgeolli*. The isolated strain was identified as a diploid *S. cerevisiae* strain by ploidy analysis using flow cytometry and named the wild-type *S. cerevisiae* GRL6. To develop this strain as a starter culture strain reducing EC formation in alcoholic beverages, the CRISPR/Cas9 genome engineering tool was utilized for multiple gene deletions in the yeast genome. The *CAR1* and *GZF3* genes were selected for gene deletion based on previous studies and successfully deleted in the genome of the *S. cerevisiae* GRL6 strain by the Cas9 for the first time. The effect of *CAR1* and *GZF3* double deletion in *S. cerevisiae* was validated by the RT-qPCR analysis which confirmed upregulation of the DUR1, 2, and DUR3 expression without the CAR1 expression. Batch fermentation of the engineered *S. cerevisiae dCAR1*, *dGZF3*, and *dCAR1*&*GZF3* in yeast growth medium containing urea and arginine showed a significant reduction of EC concentration compared to the wild-type *S. cerevisiae* GRL6 without affecting fermentation capability of strains. The double deletion strain *dCAR1*&*GZF3* decreased the most by 52.1%. The EC reduction in *Makgeolli* was also demonstrated by brewing with the engineered *S. cerevisiae*. The double deletion strain *dCAR1*&*GZF3* significantly decreased the EC concentration up to 41.6% after 168 h of fermentation without affecting ethanol concentration. This study successfully demonstrated the method of reducing EC concentration in alcoholic beverages by genome engineering of the wild *S. cerevisiae* starter culture using the CRISPR/Cas9 system. The engineered *S. cerevisiae* strains constructed in this study can be applied to reduce the EC concentration in other alcoholic beverages such as fruits wine.

## Figures and Tables

**Figure 1 foods-12-00102-f001:**
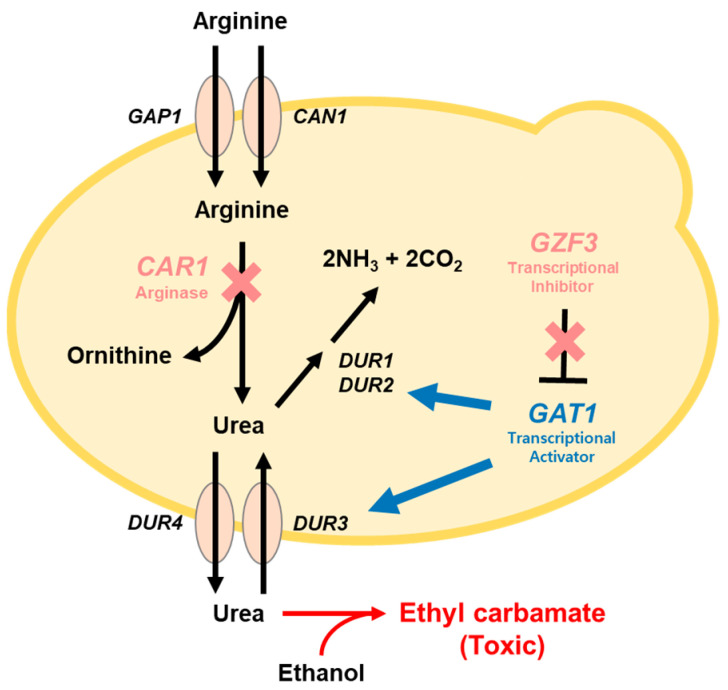
Metabolic pathway of ethyl carbamate formation from ethanol and urea in *Saccharomyces cerevisiae. S. cerevisiae* decomposes arginine into ornithine and urea by arginase enzyme encoded by *CAR1* gene. The *GZF3* gene is transcriptional activator related to the expression of urea absorption and degradation genes. *GAP1*: General amino acid permease, *CAN1*: Plasma membrane arginine permease, *CAR1*: Arginase, *DUR1*, *2*: Urea amidolyase, *DUR3*: Plasma membrane urea active transporter, *DUR4*: Urea-facilitated diffusion permease, *GZF3*: GATA zinc finger protein, *GAT1*: Transcriptional activator of nitrogen catabolite repression.

**Figure 2 foods-12-00102-f002:**
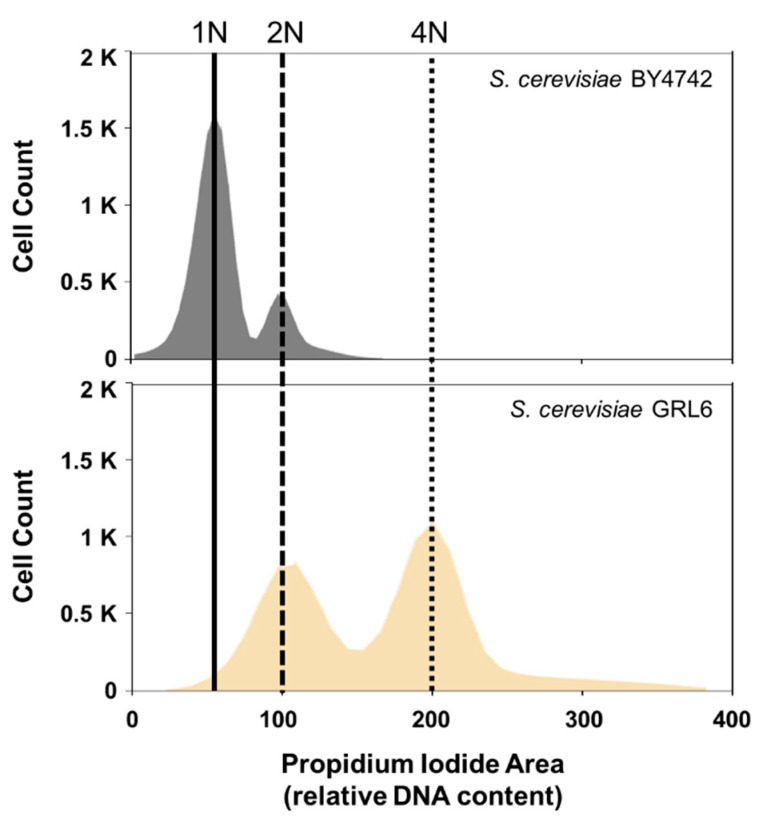
Determination of strain ploidy of industrial yeast isolated in this study. Nonparametric histogram shows the distribution of stained yeast cells according to their relative DNA content as examined by flow cytometry with propidium iodide (PI) dye using Texas Red filter. The *S. cerevisiae* BY4742 laboratory haploid strain was used as a reference strain. 1 N (haploid) *S. cerevisiae* BY4742 histogram for comparison are included at the upper part of the figure.

**Figure 3 foods-12-00102-f003:**
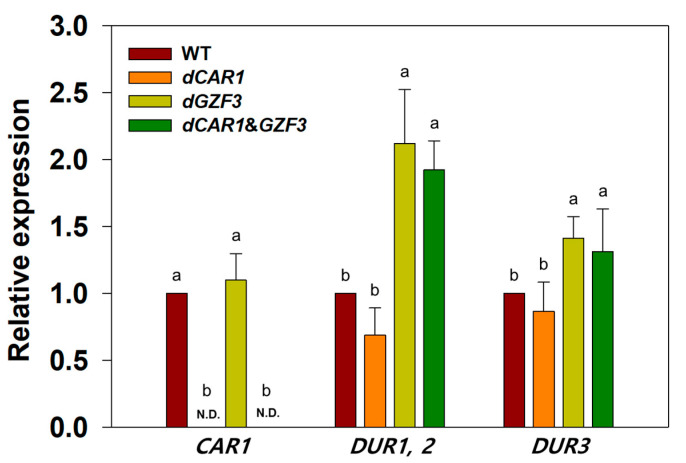
Determination of *CAR1*, *DUR1*, *2* and *DUR3* gene expression levels in the wild-type *S. cerevisiae* GRL6, and the engineered *S. cerevisiae dCAR1, dGZF3,* and *dCAR1*&*GZF3* strains grown in the YPD8 medium with 50 mg/L urea and 50 mM arginine at 30 °C for 6 h. The *ACT1* gene was used as a reference for RT-qPCR analysis using the ΔΔCt method. Data were analyzed using One-way ANOVA to determine whether the expression levels of each gene in four strains are any statistically significant differences (a and b) between the means of the groups. The *p*-values of less than 0.05 were considered statistically significant. N.D.: Not Detected.

**Figure 4 foods-12-00102-f004:**
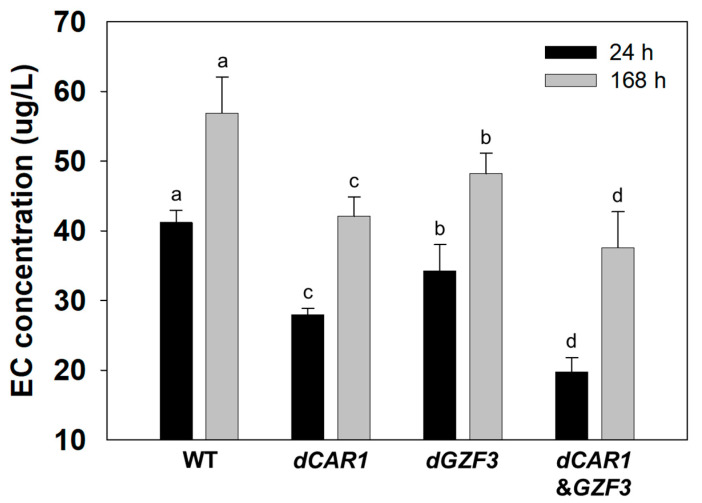
Comparison of EC concentrations in batch fermentations by the wild-type *S. cerevisiae* GRL6, and the engineered *S. cerevisiae dCAR1*, *dGZF3*, and *dCAR1*&*GZF3* strains in the YPD8 medium containing 80 g/L glucose with 50 mg/L urea and 50 mM arginine. Data were analyzed using One-way ANOVA to determine whether the EC concentrations by four strains are any statistically significant differences (a, b, c, and d) between the means of the groups at each time point (24 h or 168 h).

**Figure 5 foods-12-00102-f005:**
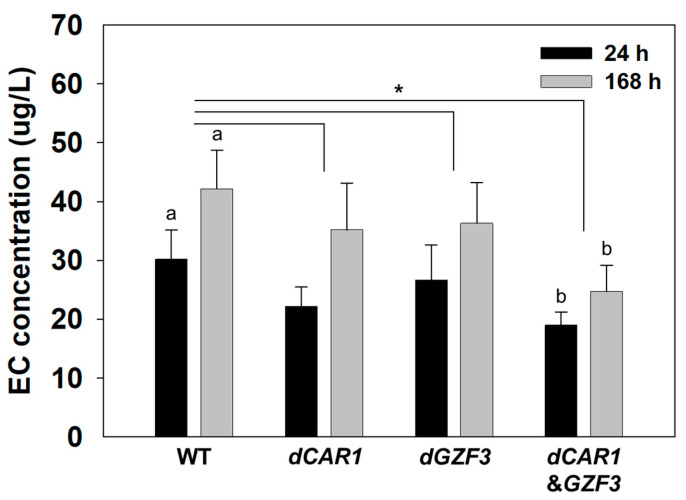
Comparison of EC concentrations in *Makgeolli* products brewed by the wild-type *S. cerevisiae* GRL6, and the engineered *S. cerevisiae dCAR1*, *dGZF3*, and *dCAR1*&*GZF3* strains. Data were analyzed using One-way ANOVA to determine whether the EC concentrations in *Makgeolli* by the engineered strains are any statistically significant differences (a and b) when compared to the wild-type strain at each time point (24 h or 168 h). * means *p* < 0.05.

**Table 1 foods-12-00102-t001:** Strains and plasmids used in this study.

Strains/Plasmids	Description	Reference/Source
Strains		
*S. cerevisiae* BY4742	*MATα his3*-Δ*1 leu2*Δ*0 lys2*Δ*0 ura3*-Δ*0*	EUROSCARF
*S. cerevisiae* GRL6	Industrial diploid yeast strain isolated from *Nuruk*	This study
Δ*CAR1*	*S. cerevisiae* GRL6 Δ*CAR1*	This study
Δ*GZF3*	*S. cerevisiae* GRL6 Δ*GZF3*	This study
Δ*CAR1*&*GZF3*	*S. cerevisiae* GRL6 Δ*CAR1* Δ*GZF3*	This study
*E. coli* TOP10	F− *mcr*A Δ(*mrr*-*hsd*RMS-*mcr*BC) ϕ80lacZΔM15 ΔlacX74 recA1 araD139 Δ(ara-*leu*)7697 *galU galK rpsL* (StrR) *endA1 nupG*	Invitrogen (CA, USA)
Plasmids		
pRS42K-gRNA-*HXK2*	*HXK2* guide RNA expression plasmid, Geneticin marker	[8]
pRS42K-gRNA-*CAR1*	Replacement of gRNA cassette for *HXK2* with gRNA cassette for deletion of *CAR1*	This study
pRS42K-gRNA-*GZF3*	Replacement of gRNA cassette for *HXK2* with gRNA cassette for deletion of *GZF3*	This study
pCas9-NAT	Cas9 expression plasmid, Nourseothricin marker	Addgene (#43802)
pRS42H	Yeast expression plasmid, Hygromycin B marker	EUROSCARF
pCas9-Hyg	Replacement of Nourseothricin marker with Hygromycin B marker	This study

**Table 2 foods-12-00102-t002:** Primers used in this study.

Name	Primer Sequence 5′→3′
Gene work	
ITS1	TCCGTAGGTGAACCTGCGG
ITS4	TCCTCCGCTTATTGATATGC
SDM-gRNA-*CAR1*-F	ggtttgaacaGTTTTAGAGCTAGAAATAGC
SDM-gRNA-*CAR1*-R	cattaggaatGATCATTTATCTTTCACTGC
SDM-gRNA-*GZF3*-F	gaacgtaagtGATCATTTATCTTTCACTG
SDM-gRNA-*GZF3*-R	ccaagttataGTTTTAGAGCTAGAAATAG
Hyg-insert-F	*ATCGACAGAGATTGTACT*GAGAGTGCAG
Hyg-insert-R	*CGGCCAGCCTCCTTA*CGCATCTGTGC
Cas9-NAT-backbone-F	*TAAGGAGGCTGGCCG*GGTGACCCGG
Cas9-NAT-backbone-R	*AGTACAATCTCTGTCGAT*TCGATACTAACGCCGCCATCC
Donor-*CAR1*-F	GAAACAACAACAACAACTATATCAATAACAATAACTACTATCAAGTTTATATCATCATCC
Donor-*CAR1*-R	ATAAAAAGAGAATGCTTATTTTGATAAAAGGGATGATGATATAAACTTGATAGTAGTTAT
Donor-*GZF3*-F	GAGAACATATTGCAAGCGGTTGAAGCTATAATACTAGATATACGAATGTATGCATATATA
Donor-*GZF3*-R	GTTTTGCAACTGATTATGCTACTATGTATTTATATATGCATACATTCGTATATCTAGTAT
Check-*CAR1*-F	CATCAGGGTTATGAGCC
Check-*CAR1*-R	GGATAACGTACCAGTGG
Check-*GZF3*-F	AGCTCGTTCCCGTCA
Check-*GZF3*-R	CTGCTTTAGTAAAAATCAAT
RT-qPCR	
*DUR1, 2*-F	GGTGTCCCTATTGCTGTTAAG
*DUR1, 2*-R	CCGTGTGCCGACTAATCC
*DUR3*-F	ACTGCCTGTGGGTGTTGTTG
*DUR3*-R	CGTCTACTGGATGCCTCTTGG
*CAR1*-F	GCTGTCCCGTGTCATTCC
*CAR1*-R	GACCTTCACCGTTTGTTTCTG
*ACT1*-F	TTATTGATAACGGTTCTGGTATG
*ACT1*-R	CCTTGGTGTCTTGGTCTAC

The italic sequences present the homologous region for Gibson assembly. The small letter sequences present the replacement site region for Site Directed Mutagenesis. The underlined sequences present the homologous region for overlap PCR.

## Data Availability

Data is contained within the article or Appendix A.

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
