# Peer review of "Reduction of Ethyl Carbamate in an Alcoholic Beverage by CRISPR/Cas9-Based Genome Editing of the Wild Yeast"

_foods, 2022, doi:10.3390/foods12010102_

Round 1
Reviewer 1 Report
The manuscript mainly describes the reduction of ethyl carbamate (EC) in alcoholic beverages using genetic engineering S. cerevisiae based on the CRISPR/Cas9 strategy. Four strains, including two with single gene mutation, one with double gene mutation, and wild type, were tested for their ethanol and EC production. The authors concluded that the deletion of CAR1 and GZF3 significantly reduced EC formation without affecting ethanol production. The content in this manuscript is well-organized. However, the authors should clarify some points.
1) The authors mentioned that EC is a carcinogen and consumption of EC may cause severe problems to human health. It is better to include more information about the quantity of EC in alcoholic beverages and the amount of EC that has a negative effect on human health.
2) There are several metabolic engineering techniques for genetic modification in the microbial cell; thus, it is better to provide some information regarding this point, the pros and cons of such methods, and the advantage of using CRISPR/Cas9 technology in this study in the introduction.
3) Table 2, please provide the expected size of the PCR product after using a different set of primers. Furthermore, the paragraph "The underlined sequences present the homologous region for overlap PCR" does not match the data presented in this table; please recheck.
4) The authors should provide the conditions of RT-qPCR in the Materials and methods.
5) For Korean rice wine production, the authors should provide the yeast cell number instead of OD600 in the Materials and method.
6) In the results and discussion, the authors mentioned the isolation of S. cerevisiae from Nuruk. It is better to provide more information about the number of yeast isolated obtained from Nuruk and the characteristics of S. cerevisiae GRL6, which was selected to use in this study.
7) S. cerevisiae strain dGZF3 still produces EC content higher than strain dCAR1; please elaborate on these results (why strain dGZF3 had a higher level of EC than dCAR1).
8) The statistical analysis and the description of the statistical results in each Figure (Fig. 3, 4, and 5) should be clearly described.
9) In Supplementary Tables S2 and S3, the EC concentrations at 0 h were slightly high. The authors should provide and describe this point.
Reviewer 2 Report
In this study, the authors successfully isolated the wild type of S. cerevisiae from traditional Korean rice wine and developed a starter culture strain to reduce the formation of EC in alcoholic beverages by genome engineering the wild S. cerevisiae starter culture using the CRISPR/Cas9 system. I suggest that the authors refer to the regulatory requirements and the use of these strains in the industrial process in the introduction.
